# Preparation and Application of a Urea–Formaldehyde-Blended Guanidinium Azole–Phytic Acid–Copper Flame-Retardant Resin Coating

**DOI:** 10.3390/polym16233366

**Published:** 2024-11-29

**Authors:** Xulan Lu, An Wei, Shunxiang Wang, Yongjing Zou, Yunhao Lu, Lixian Sun, Cuili Xiang

**Affiliations:** 1School of Materials Science and Engineering, Guilin University of Electronic Science and Technology, Guilin 541004, Chinasunlx@guet.edu.cn (L.S.); 2Nanning Guitian Electronic Technology Research Institute Co., Ltd., Nanning 530000, China; wa123@mails.guet.edu.cn (A.W.);; 3School of Materials Science and Engineering, Zhejiang University, Hangzhou 310027, China; luyh@zju.edu.cn

**Keywords:** flame retardant, urea–formaldehyde resin, coating

## Abstract

In this study, environmentally friendly flame retardants capable of efficient flame retardancy at low concentrations in wood were developed. Urea-formaldehyde (UF) resin and guanidinium azole (GZ)-phytate (PA)-copper hydroxide (Cu(OH)_2_) flame-retardant resin coating blends were prepared using urea, formaldehyde, 3,5-diamino-1,2,4-triazole (GZ), phytanic acid (PA), and copper hydroxide (Cu(OH)_2_). Employing dioctyl phthalate as the plasticizer and tannic acid as the curing agent, a three-stage reaction was performed to obtain the desired UF-GZ/PA/Cu as a bio-based flame retardant. Thermal evaluations demonstrated that UF-GZ/PA/Cu lost 5% of its mass through decomposition at a temperature of 195.5 ± 2.1 °C, with its maximum decomposition rate being observed at 300.6 ± 1.5 °C, and 29.8 ± 2.5 wt.% of dense residual charcoal being obtained at 800 °C. When applied as a flame retardant coating on wood, the prepared UF-GZ/PA/Cu exhibited excellent flame-retardant properties, forming a continuous dense charcoal residual layer, with a limiting oxygen index of 32.0%, and passing the UL-94 V-0 test. In addition, the heat release rate and total heat release rate of the flame retardant were determined to be reduced by 87.7 and 83.66%, respectively. Overall, this study provides a green and effective method for the preparation of flame-retardant wood.

## 1. Introduction

Wood is a common everyday item in construction, fabrication, military, and energy applications, among others [1,2,3,4]. However, the high flammability of wood results in great losses of life and material every year [5,6]. The development of flame-retardant wood can be achieved by impregnating wood with a flame-retardant solution or by applying a coating on the wood surface [7,8]. For example, in the case of flame-retardant bonded-coated wood, a flame-retardant agent is physically mixed or chemically reacted to create a coating that is applied to the wood surface [1,9,10].

Halogenated flame retardants have been banned by various countries, and so the development of low-toxicity, environmentally friendly, halogen-free flame retardants has received growing attention in recent years [11,12]. Intumescent flame retardants (IFRs) are generally formed by foaming and expanding phosphorus-based compounds (as acid sources), high-carbon polyhydroxy compounds (as carbon sources), and nitrogen-containing compounds (as gas sources) [13,14,15,16]. Phytic acid (PA), which is a green raw material that is extracted from plants, is rich in phosphorus and can be chelated with metals to form phytic acid metal salts [17,18,19]. In this context, Tang et al. [20] incorporated PA-Na into an intermediate (V-Cc) generated by the reaction of melamine and vanillin with 1,4 dioxane and polyethyleneimine (PEI). The resulting flame retardant was added to epoxy (EP) resin with DDM and then cured to obtain the desired EP-V-Cc-PP composite, which self-extinguished with an ultimate oxygen index of 27.8%. In another study, Yu et al. [21] synthesized a lignin-based P-N flame retardant. More specifically, urea, PA, and carboxymethyl lignin were reacted to form a phytate–urea–carboxymethyl lignin, named PUC. It was demonstrated that polyphosphates from the PA breakdown, while lignin promotes dehydration to form a dense charcoal layer, resulting in a 56.8% reduction in the total heat released (THR) and a 92.3% reduction in the total smoke production (TSP). This system therefore exhibited significantly improved flame-retardant properties.

Commonly used flame-retardant coatings include EP resins, glycol resins, vinyl ester resins, and urea-formaldehyde (UF) resins [22,23,24]. For example, Peng et al. [25] mixed 9,10-dihydro-9-oxa-10-phosphophenanthrene-10-oxide (DOPO) and bis-phthalonitrile to produce the yellow flame-retardant compound PBD in a one-pot method. The prepared PBD was then added to phenolic resin to generate a flame-retardant coating. After curing, the flame-retardant EP formed a semi-interpenetrating network and a hyperbranched structure. It was demonstrated that this EP composite exhibited a limiting oxygen index (LOI) of 49.5%. In addition, the residual carbon content of PBD-20 reached 37.3% at 700 °C, thereby demonstrating the good flame-retardant properties of the EP composite.

Guanidinium azole (GZ) is a yellow, nitrogen-rich, crystalline powder [26,27]. GZ not only is a flame retardant but also adsorbs carbon dioxide, thereby reducing the amount of smoke released during combustion. GZ can also form complexes with metals to form guanidazole salts, leading to its application as an excitatory drug in medicine [28]. In a previous study, Yang et al. [26] prepared flame-retardant coatings of GZ, tris(hydroxymethyl)phosphine oxide (THPO), and polydimethylsiloxane (PTMS) to impart excellent flame-retardant properties to cotton fabrics; however, the costs associated with these raw materials were extremely high.

In this study, GZ/PA/metal flame retardants are prepared via a one-pot approach (Figure 1). Initially, a GZ/PA solution is formed by a simple complexation reaction between GZ and PA, and subsequently the GZ/PA/Cu flame retardant is produced via chelation with Cu(OH)_2_. In addition, a UF adhesive coating is synthesized by the addition of GZ/PA/Cu to dihydroxymethylurea with homogeneous mixing to produce a UF-GZ/PA/Cu coating. Subsequently, GZ/PA/Mg, GZ/PA/Ca, and GZ/PA/Mn flame retardants are prepared by substituting Cu with Mg, Ca, and Mn, respectively. The corresponding UF-GZ/PA/Mg, UF-GZ/PA/Ca, and UF-GZ/PA/Mn coatings are also prepared.

## 2. Materials and Methods

### 2.1. Materials

Formaldehyde solution (37 wt.% aqueous solution), PA (70 wt.% aqueous solution), GZ, Cu(OH)_2_, Mg(OH)_2_, Ca(OH)_2_, MnO_2_, and urea (AR) were purchased from Xilong Reagent (Shenzhen, China). Dioctyl phthalate (DOP, AR) was purchased from Aladdin Reagent Co., Ltd. (Shanghai, China). Tannic acid (TA, AR), NaOH (20 wt.% aqueous solution), and acetic acid (CH_3_COOH, 15 wt.% aqueous solution) were prepared in the laboratory.

### 2.2. Synthesis of the Flame-Retardant GZ/PA/Metal Species

GZ (4.459 g) was dissolved in deionized water (150 mL) in a three-necked flask under stirring at 600 rpm and heated to 70 °C in a water bath. A 37 wt.% PA solution (14.25 mL) was then added dropwise to the stirring GZ solution at a rate of 0.5 drop/s. Complete addition was achieved after 30 min, after which point the temperature was raised to 90 °C, Cu(OH)_2_ (8.664 g) was added, and the mixture was allowed to react for 3 h. After this time, the product was filtered and washed with deionized water until the pH of the filtrate reached 5–6. The filter slag was then dried at 80 °C for 24 h and ground to give the desired GZ/PA/Cu as a powder. In addition, the reaction was repeated by replacing Cu(OH)_2_ with Mg(OH)_2_ (3.954 g), Ca(OH)_2_ (5.024 g), or MgO_2_ (3.951 g) to prepare the corresponding GZ/PA/Mg, GZ/PA/Ca, and GZ/PA/Mn specimens, respectively.

### 2.3. Preparation of the Modified UF

A 20 wt.% NaOH solution was first added to a 37 wt.% formaldehyde solution (100 g); the solution pH was adjusted to 8.0–8.5, and the mixture was heated to 90 °C under 400 rpm stirring. Subsequently, the modified UF resin emulsion was prepared by adding urea in three stages. First, urea (37 g) was added at 90 °C and stirred for 30 min, then the desired volume of a 20% acetic acid solution was added to adjust the pH of the solution to 4.5–5.0. Urea (12.4 g) and the above-prepared GZ/PA/Cu (1 g) were then added and stirred at 90 °C until the reaction end point was reached (i.e., when a drop of the reaction solution did not disperse when added to water at 30 °C). In the third stage, maintaining a temperature of 90 °C, a 20 wt.% sodium hydroxide solution was added to adjust the solution pH to 7.5–8.0. Subsequently, urea (7.6 g) was added, the temperature was adjusted to 70 °C, and the modified UF was obtained by stirring for 30 min.

To obtain the UF-GZ/PA/Cu flame-retardant coating [29], DOP (1 mL) and TA (1.2 g) were added to the above mixture under stirring at 400 rpm for 30 min.

The above processes were repeated using the GZ/PA/Mg, GZ/PA/Ca, and GZ/PA/Mn specimens. They are outlined schematically in Figure 2.

### 2.4. Characterization

The micromorphology and surface elements of the flame retardants were characterized using scanning electron microscopy (SEM, Quanta 450, EFG, FEI, Hillsboro, OR, USA) and energy-dispersive X-ray spectroscopy (EDS). For SEM, the surface morphology was observed at an accelerating voltage of 5 kV and an electron intensity of 8 eV. To increase the signal strength during EDS analysis, the accelerating voltage was increased to 10 kV, and the electron intensity was increased to 9 eV. The degree of graphitization of the UF residues was analyzed using Raman spectroscopy (HORIBA Scientific, Paris, France) at a laser excitation wavelength of 532 nm. Fourier-transform infrared (FTIR, Nicolet 6700, Thermo Scientific, Waltham, MA, USA) spectroscopy was performed between 4000 and 500 cm^−1^. All materials were also examined using X-ray diffractometry (XRD, D8 Advance, Bruker, Germany) under Cu Kα radiation conditions at 45 kV and 200 mA with a scan rate of 0.1° s^−1^. X-ray photoelectron spectroscopy (XPS, Thermo Fisher Scientific, Waltham, MA, USA) was used to characterize the surface functionalities and chemical compositions of the samples.

The UF-GZ/PA/metal species were analyzed using thermogravimetric analysis (TGA, Thermo Fisher Scientific, Waltham, MA, USA) with a test atmosphere of nitrogen, a temperature range of 50–800 °C, and a ramp rate of 20.0 °C min^−1^. A UL-94 vertical flame test was conducted (CZF-3 vertical flame tester, Jiangning Analytical Instruments Co., Ltd., Nanjing, China) using samples with dimensions of 130 mm × 13 mm × 3 mm. The extreme oxygen index tests (HC-2 oxygen index tester, Jiangning Analytical Instruments Co., Ltd., Nanjing, China) were performed using samples with dimensions of 130 mm × 6.5 mm × 3 mm. Cone calorimetry tests (TTech-GBT161172 cone calorimeter, Testech Technology, Suzhou, China) were conducted using samples with dimensions of 100 mm × 100 mm × 3 mm. The curing behaviors of the materials were analyzed by differential scanning calorimetry (DSC, DSC-250, TA Instruments, Newcastle, DE, USA) under a N_2_ atmosphere. The DSC experiments were performed over a temperature range of 40–160 °C and with a heating rate of 10 °C min^−1^. The impact strengths of the materials were tested using a simple supported beam impact tester (Shanghai Hesheng Instrument Technology Co., Ltd., Shanghai, China).

## 3. Results and Discussion

### 3.1. Characterization of the Synthesized GZ/PA/Metal Species

XRD analysis was performed to determine the composition of the prepared GZ/PA/Cu. As shown in Figure 3a, characteristic peaks are present for Cu_4_O(PO_4_)_2_ and CuO, confirming the successful chelation of Cu with PA. Similarly, the XRD results for GZ/PA/Mg indicated the presence of Mg_3_P_2_ and Mg_2_P_2_O_7_; a peak corresponding to Ca(PO_3_)_2_ was observed for GZ/PA/Ca, and a characteristic peak of Mn(PO_3_)_3_ was detected for GZ/PA/Mn, confirming metal chelation with PA. Subsequently, the FTIR spectra of the prepared GZ/PA/Cu were analyzed as shown in Figure 3b. More specifically, an absorption signal corresponding to the –NH/–OH telescopic vibration of GZ was observed at 3434.79 cm^−1^, as was a characteristic peak corresponding to the –NH bending vibration at 1637.67 cm^−1^ [30]. In addition, peaks corresponding to the –OH vibration and the telescopic P=O vibration were observed at 1386.77 and 1075.82 cm^−1^, respectively, both of which originated from the PA component. Furthermore, broad P–O–C vibrational peaks appeared at 999.43 cm^−1^ [31,32], thereby indicating that GZ formed a complex with PA. These results confirm the successful synthesis of GZ/PA/Cu, and similar results were obtained for the other GZ/PA/metal species.

XPS was subsequently employed to investigate the chemical structures of GZ/PA/Cu and the other GZ/PA/metal species. As shown in the XPS plots recorded for GZ/PA/Cu (Figure 4a–e), the peaks at 531.78, 532.98, and 534.08 eV in the O 1s spectrum correspond to the P–O, O–C–O, and P=O moieties, respectively, whereas the 284.88, 287.28, and 289.18 eV peaks in the C 1s spectrum were attributed to the C–C/C–H, C–O–C, and C=O moieties. In addition, the peaks observed at 399.98 and 400.28 eV in the N 1s spectrum correlate with the P–N/C–N and N–H groups, while in the P2p spectrum, the peaks at 133.38 and 134.08 eV belong to the P–O–C and P=O groups, respectively [33]. These analyses confirm complexation between GZ and PA. In addition, the Cu 2p spectrum in Figure 4e showed peaks corresponding to Cu^2+^ and Cu^+^ species at 932.68 and 934.78 eV, respectively, indicating that the PA was successfully chelated with Cu [34], and confirming successful preparation of the GZ/PA/Cu flame retardant. The Mg 1s, Ca 2p, and Mn 2p spectra were also recorded for the corresponding species (see Figure 4f, Figure 4g, and Figure 4h, respectively) [35]. Among them, the Ca 2p spectrum was deconvoluted into Ca2p_1/2_ and Ca2p_3/2_ peaks, while the Mn2p spectrum was deconvoluted into Mn 2p_3/2_ and Mn 2p_1/2_ peaks [36]. These results indicate that the GZ/PA/Mg, GZ/PA/Ca, and GZ/PA/Mn flame retardants were successfully prepared.

To examine the morphologies and chemical compositions of the prepared GZ/PA/Cu, GZ/PA/Mg, GZ/PA/Ca, and GZ/PA/Mn species, SEM and EDS were employed. As shown in Figure 5a, the microscopic GZ/PA/Cu image shows a fluffy, flocculent structure, which was confirmed by EDS to contain N, P, and Cu, confirming that this structure was the GZ/PA/Cu flame retardant. Similarly, Figure 5b shows the microscopic morphology of GZ/PA/Mg, which exhibits irregular granules. EDS demonstrated that these agglomerations contain N, P, and Mg, indicating the successful synthesis of GZ/PA/Mg. Furthermore, Figure 5c,d show the morphologies of GZ/PA/Mn and GZ/PA/Ca, which exist as irregular and unevenly distributed agglomerations. The presence of metallic Ca was confirmed for GZ/PA/Ca, while Mn was detected for GZ/PA/Mn, thereby confirming the successful preparation of these species.

### 3.2. Combustion and Flame-Retardant Properties of the Prepared Samples

The LOI and vertical burning (UL-94) tests are commonly used methods for evaluating the flame retardancy properties of materials. The results of the LOI tests for the various materials are presented in Table 1 and Figure 6, wherein it can be seen that pure wood (NW) was flammable, with an LOI of only 20.3%. Notably, the LOI increased to 27.8% when coated with UF and increased again to 32.0% in the presence of the UF-GZ/PA/Cu flame-retardant coating. Considering the various GZ/PA/metal systems employed, UF-GZ/PA/Cu > UF-GZ/PA/Mg > UF-GZ/PA/Ca > UF-GZ/PA/Mn > UF > NW, which suggests that the addition of a UF-GZ/PA/metal flame retardant can effectively enhance the flame-retardant properties of the NW, with the optimal flame-retardant properties being observed for UF-GZ-PA/Cu. This can be attributed to the fact that Cu exhibited a superior dispersion than the other metals, which is conducive to improved flame retardancy. In the vertical combustion test, the first ignition time of NW was determined to be >60 s, which does not meet the UL-94 rating standard. Notably, the UF-coated wood achieved the UL-94-V-0 rating, and those coated with UF-GZ/PA/Cu, UF-GZ/PA/Mg, UF-GZ/PA/Ca, and UF-GZ/PA/Mn met the UL-94-V-0 rating. While the various metals were able to meet the V-0 test, the time to first ignition for the UF-GZ/PA/Cu sample was only 0.9 s, with no dripping observed during combustion; this flame retardant therefore exhibited the best flame retardant performance of the various samples evaluated.

To evaluate the flame-retardant properties of the modified wood, a cone calorimetry test (CCT) was performed to simulate the burning of an actual flame. The flame-retardant properties of the modified wood were confirmed, as shown in Figure 6 and Table 2, wherein the time to ignition (TTI), heat release rate (HRR), total heat released (THR), average effective heat of combustion (av-EHC), CO production rate (COP), and total mass loss rate (TML) are provided. More specifically, as shown in Table 2, the TTI of the NW is 136 s, which remained unchanged after coating with UF. However, the ignition time was shorter upon coating with the UF-GZ/PA/Cu flame retardant. It was also found that the PA component of the UF-GZ/PA/Cu decomposed at an early stage to form phosphite and/or orthophosphate, and this degradation was promoted by the presence of Cu, ultimately reducing the ignition time by 3 s. Notably, the TTI was significantly shortened for the UF-GZ/PA/Mg, UF-GZ/PA/Mn, and UF-GZ/PA/Ca coatings. Furthermore, as presented in Figure 6a, the HRR of NW reached 206.73 kW m^−2^, and this was reduced to 87.22 kW m^−2^ in the presence of the UF coating. Importantly, the addition of the GZ/PA/metal species further reduced the HRR, with the most significant reduction being observed for GZ/PA/Cu, that is, an HRR of 25.38 kW m^−2^, which represents an 87.7% reduction. Moreover, the THR of the NW was 21.31 kW m^−2^, while that recorded in the presence of the UF-GZ/PA/Cu coating was only 3.48 kW m^−2^ (Figure 6b). In addition, from Figure 6c, it can be seen that the mass loss curve of UF-GZ/PA/Cu exhibits the lowest mass loss of 78.3 wt.% after combustion. Considering that smoke emissions and CO production are the main causes of human death due to fire, the COP rates were evaluated for the various systems, as presented in Figure 6d and Table 2. More specifically, NW produces CO at a rate of 0.0024 g s^−1^, while GZ/PA/Cu produces CO at a significantly lower rate of 0.00006 g s^−1^. This can be attributed to the fact that, after combustion, GZ/PA/Cu thermally decomposes the metal oxides, which consume oxygen and cause the material to exhibit a self-extinguishing behavior and reduce the amount of CO. These results therefore clearly demonstrate that the addition of a UF-GZ/PA/Cu coating significantly enhances the flame retardancy of wood and reduces the amount of CO produced.

After analyzing by the above method, it can be seen that UF-GZ/PA/Cu has the best flame-retardant performance. In order to highlight the flame-retardant performance characteristics of UF-GZ-PA/Cu flame retardants with the research based on phytate metal-based flame retardants in recent years, they were compared based on the four flame retardants, namely, melamine phytate (MPA), melamine phytate copper (MPACu), melamine phytate zinc (MPAZn), and melamine phytate magnesium (MPAMg), which were prepared by Dong et al. [37], as shown in Table 3. After analyzing and comparing their data, it can be seen that MPACu’s has little effect on improving the flame-retardant properties of the matrix. And MPAZn has the best flame-retardant performance. When the addition amount was 20 wt.%, the LOI was 25.0%, the HRR and THR were reduced by 27.5% and 21.0%, respectively, and the amount of residual carbon at 800 °C was only 15 wt.%. Comparison with the flame-retardant properties of the UF-GZ/PA/Cu flame-retardant coatings of the present study clearly shows that the addition of GZ/PA/Cu is only 1 wt.%, and the LOI value is 32.0%, which is an improvement of 28.0%. And the improvements of HRR and THR were 87.7% and 83.6%, respectively. The amount of residual carbon reached 29.8 wt.%. The data analysis showed that the flame-retardant properties of UF-GZ/PA/Cu were significantly better than those of MPAZn. And the addition amount is obviously much less than that of MPAZn.

To further highlight the flame-retardant performance characteristics of UF-GZ/PA/Cu, WPU composites based on Cu(II)-phytate-aromatic Schiff base complexes of aromatic Schiff base derivatives were prepared by Gan et al. [38]. The results show an LOI of only 27.5%, an increase in HRR instead of a decrease, and a decrease in THR of only about 18.0%. The residual carbon at 600 °C is 8.5 wt.%.

In summary, after the comparison of MPAZn and Cu(Ⅱ)-PA/WPU-HIMB with UF-GZ/PA/Cu, it can be seen that UF-GZ/PA/Cu flame-retardant coatings have the best flame-retardant effect, and the amount of GZ/PA/Cu flame retardant is only 1%, which means that a small amount of flame retardant can be added to obtain a better flame-retardant effect, and the product has rich sources of raw materials and low cost, which is conducive to the realization of the factory. It is beneficial to realize factory production.

### 3.3. Analysis of the Flame-Retardant Mechanism

To further understand the flame retardancy behavior of GZ/PA/Cu, a range of analytical techniques were employed (i.e., SEM, EDS, Raman spectroscopy, and XPS) to characterize the amount of residual carbon after combustion. For this purpose, cone calorimetric tests were performed using UF, UF-GZ/PA/Cu, UF-GZ/PA/Mg, UF-GZ/PA/Ca, and UF-GZ/PA/Mn, and the residual carbon present after combustion is shown in the photographic images presented in Figure 7a. It can be seen that the specimens coated with UF and UF-GZ/PA/metal were slightly darker in color than the NW. NW was found to be completely burned with very little charcoal ash remaining, while the burned expanded charcoal layer after coating with the UF flame retardant could reach 2.00 cm, and the burned expanded charcoal layers after coating with UF-GZ/PA/Cu, UF-GZ/PA/Mg, UF-GZ/PA/Ca, and UF-GZ/PA/Mn reached 4.10, 3.10, 2.15, and 2.40 cm, respectively. The burned residual char of the UF-GZ/PA/Mg system was then subjected to SEM and EDS, as shown in Figure 7b. The presence of Mg, P, and N on the carbon layer surface was confirmed by EDS, indicating that the UF-GZ/PA/Mg was successfully adhered to the wood. A small number of holes existed in this carbon layer due to gas release during heating. Figure 7c presents the corresponding results for the UF-GZ/PA/Mn system, which also contains some pores on its surface, while Figure 7d shows slight fragmentation of the charcoal layer and the presence of pores for the UF-GZ/PA/Ca system. These observations indicate that the carbon layers formed by chelating Mg, Mn, and Ca with PA cannot effectively resist the effects of gas release and that the quality of the carbon layer must be improved to produce a general flame-retardant effect. In contrast, Figure 7e shows the microscopic morphology of the residual carbon of the UF-GZ/PA/Cu system, from which it can be seen that the surface layer of carbon is dense due to the homogeneous dispersion of Cu in the UF-GZ/PA solution, leading to an improved dispersion over the substrate and an increased contact area. Furthermore, under heating, Cu forms oxide fillers on the carbon layer surface, producing a denser layer and effectively insulating against heat and oxygen to produce a superior flame-retardant effect. The results demonstrate that the UF coating was not effective in terms of improving the quality of the carbon layer in NW, while the addition of UF-GZ/PA/Cu was beneficial. This can be attributed to the fact that generated radical nitrogen species in GZ can combine with oxygen radicals to form a non-combustible gas. In addition, PA decomposes under heating to capture oxygen radicals and generate polyphosphate, which dehydrates to promote residual char formation. Consequently, the escape of combustible gases is prevented, the wood is protected from the air, and the Cu ions combine with O ions to form oxides that are deposited on the carbon layer; the synergistic effect between these oxides and PA promotes the formation of residues.

The TGA and differential thermogravimetric (DTG) curves recorded for the UF and UF-GZ/PA/metal species under an N_2_ atmosphere are shown in Figure 8, and the detailed data are summarized in Table 4. It can be seen that the UF had lost 5% of its mass through decomposition at 204.2 °C. In addition, the temperature at which its maximum decomposition rate was 304.2 °C and the amount of residual carbon at 800 °C was 10.0 wt.%. The 5% decomposition temperatures of UF-GZ/PA/Cu, UF-GZ/PA/Ca, and UF-GZ/PA/Mn were determined to be 195.5, 191.4, and 184.0 °C, respectively, representing reductions of 8.7, 12.5, and 20.2 °C compared to that of the UF system. In contrast, the 5% decomposition temperature of UF-GZ/PA/Mg was increased by 1.0 °C to 205.2 °C. Furthermore, the UF-GZ/PA/Cu, UF-GZ/PA/Mg, UF-GZ/PA/Ca, and UF-GZ/PA/Mn specimens reached their maximum decomposition rates at 300.6, 302.8, 301.5, and 304.1 °C, representing decreases of 3.6, 1.4, 3.1, and 0.1 °C, respectively, compared to the UF system. Moreover, the char residue contents of UF-GZ/PA/Cu, UF-GZ/PA/Mg, UF-GZ/PA/Ca, and UF-GZ/PA/Mn at 800.0 °C were 29.8, 27.3, 29.4, and 26.0 wt.%, respectively, representing increases of 19.8, 17.3, 19.4, and 16.0 wt.% compared to that of the UF system. These observations suggest that the decomposition of PA in the UF-GZ/PA/metal species produces phosphite and/or orthophosphate in early stages, and the metal ions promote the early decomposition of the material [39]. It was also evident that a more stable carbon residue could be produced in the high-temperature region in the presence of these metal ions due to their ability to form peroxides and oxides during decomposition. Furthermore, differences were observed between the flame-retardant effects of the four metals, with the UF-GZ/PA/Cu system giving an optimal result. This was attributed to the coupling of the PA molecular orbitals with the metal orbitals through delocalization of the metal center. More specifically, Cu belongs to the group IB transition metals, which are characterized by high-temperature resistance. In contrast, the group IIA elements Mg and Ca are more active than Cu, with the most stable PA complex forming between Cu^2+^ and PA. While Mn belongs to the group ⅦB transition metal elements, the complex formed between Mn^4+^ and PA is unstable. In the context of compatibility, the formation of a more uniformly dispersed mixture leads to superior compatibility, such as that observed in the case of Cu, as described earlier. This can also be accounted for by considering the intermolecular forces and affinity within the system, wherein stronger forces lead to a more uniform dispersion and a superior compatibility. In addition, Cu ions possess a high thermal conductivity and exist in a multivalent state after combustion, thereby promoting the consumption of oxygen and enhancing the flame-retardant effect.

While the UF coating itself exhibits a degree of flame retardancy, it generates a fragmented residual carbon layer of poor quality, which limits the flame retardancy of the system. In contrast, the addition of GZ/PA/metal to the coating increases the amount of residual carbon and the quality of the carbon layer after combustion, with GZ/PA/Cu forming a dense residual carbon layer to significantly improve the flame retardancy.

Subsequently, XPS was used to determine the chemical structure of the UF-GZ/PA/Cu residual carbon after combustion. In the O 1s spectrum presented in Figure 9a, the peaks at 531.58 and 532.58 eV correspond to the P–O and O–C–O bonds, respectively, while in the P 2p spectrum (Figure 9d), the peaks at 1207.35 and 133.98 eV correspond to the P–O–C and P=O bonds, respectively. In addition, in the N 1s spectrum (Figure 9b), peaks corresponding to P–N/C–N bonds (398.78 eV) and N–H bonds (400.48 eV) are evident, while in the C 1s spectrum (Figure 9c), the peaks at 284.78, 286.18, and 288.78 eV correspond to the C–C/C–H, C–O–C, and C=O bonds, respectively [40]. These results indicate that more polyphosphates and nitrogenous compounds are generated during the combustion of UF-GZ/PA/Cu on the wood surface. Furthermore, in the Cu 2p spectrum (Figure 9e), peaks corresponding to Cu^2+^, Cu^+^, and Cu^0^ can be seen at 932.28, 934.58, and 952.9 eV, respectively [41]. This confirms the presence of various oxidized valence states of copper after combustion, which can synergistically promote charring with the polyphosphates. More specifically, Cu^2+^ is expected to bind more strongly to the functional groups of PA. Figure 9f, Figure 9g, and Figure 9h show the Mg 1s, Ca 2p, and Mn 2p spectra of UF-GZ/PA/Mg, UF-GZ/PA/Ca, and UF-GZ/PA/Mn, respectively. As shown, the presence of Mg^2+^ (1303.08 and 1304.88 eV) is confirmed in the Mg 1s spectrum, while the existence of Ca^2+^ (347.58 and 350.58 eV) is evident from the Ca 2p spectrum. Similarly, Mn^2+^ (641.48 eV) and Mn^3+^ (647.58 eV) are present in the Mn 2p spectrum [42,43]. These results confirm that the metallic elements of UF-GZ/PA/Mg, UF-GZ/PA/Ca, and UF-GZ/PA/Mn are present in the same valence form as that detected in the char residue after combustion.

Raman spectra recorded from the residual charcoal obtained from the UF and UF-GZ/PA/metal systems were used to determine the degree of graphitization (Figure 10). The ratio between the D peak (1350 cm^−1^) and the G peak (1580 cm^−1^), known as I_D_/I_G_, was used as an indication of the degree of graphitization. More specifically, a higher value of I_D_/I_G_ corresponds to denser packing of the charcoal particles, thereby imparting a greater degree of protection [44]. Therefore, according to Raman results, the I_D_/I_G_ value of the UF system was 1.39, while those of the UF-GZ/PA/Mg, UF-GZ/PA/Ca, UF-GZ/PA/Mn, and UF-GZ/PA/Cu systems were 1.64, 1.76, 1.41, and 1.95, respectively. These results indicate that the UF-GZ/PA/Cu charcoal particles exhibited the optimal charring ability and the best char layer quality.

Based on the above analysis, the mechanism responsible for the flame-retardant behaviors of the UF-GZ/PA/metal systems in gas and condensed phases was deduced (see Figure 11). More specifically, under heating, GZ captures oxygen radicals to generate NO_2_ and other non-combustible gases. In addition, the metal elements are heated and combine with oxygen radicals to generate oxides or peroxides that promote dehydrogenation and decarboxylation reactions. Furthermore, PA decomposes into polyphosphates and other substances, and ultimately, the synergistic effect between the metal elements and PA promotes the generation of residual carbon [45]. Considering that during the preparation of the GZ/PA/metal system, Cu^2+^ is mixed more homogeneously due to its superior ionic binding affinity, biocompatibility, and dispersibility, a higher thermal conductivity is achieved compared to the other metal systems. The resulting dispersion then increases the contact area with the UF to achieve optimal results, thereby accounting for the enhanced flame-retardant effect exhibited by this system.

### 3.4. Other Properties of the Modified Wood

The NW coated with the UF and UF-GZ/PA/metal species were dried for impact strength testing, as presented in Figure 12a. Consequently, the impact strengths of the NW samples coated with UF, UF-GZ/PA/Mg, UF-GZ/PA/Ca, UF-GZ/PA/Cu, and UF-GZ/PA/Mn were determined to be 11.7, 11.8, 12.0, 14.9, and 14.8 kJ m^−2^, respectively. These results demonstrate that the addition of GZ/PA/metal significantly increased the impact strength of the NW, with the highest impact strength being observed for GZ/PA/Cu.

Finally, the UF and UF-GZ/PA/metal species were analyzed by DSC, and the results are shown in Figure 12b. More specifically, exothermic peaks were observed for UF, UF-GZ/PA/Ca, UF-GZ/PA/Mn, UF-GZ/PA/Cu, and UF-GZ/PA/Mg at 74.6, 76.8, 78.3, 91.5, and 81.6 °C, respectively. These values demonstrate that the addition of GZ/PA/metal prolonged the curing time and increased the curing temperature of UF. This, in turn, can be attributed to the fact that the synthesized GZ/PA/metal acts as a macromolecule, inhibiting intermolecular movement and thereby delaying the curing time.

## 4. Conclusions

A novel urea-formaldehyde (UF)–guanidazole–phytate–copper (GZ/PA/Cu) resin flame-retardant coating was prepared via a complexation reaction. To clearly evaluate the flame-retardant properties of the UF-GZ/PA/Cu flame retardant, the copper component was replaced with other metals (i.e., Mg^2+^, Ca^2+^, and Mn^4+^). It was found that an optimal flame retardancy was achieved by adding only 1% of UF-GZ/PA/Cu to the flame-retardant coating. Thermal evaluation showed that UF-GZ/PA/Cu decomposed and lost 5% of its mass at 195.5 ± 2.1 °C, with the maximum decomposition rate being detected at 300.6 ± 1.5 °C and a residual carbon amount of 29.8 ± 2.5 wt.% being obtained at 800 °C, which corresponded to a high-quality char layer. In addition, the heat release rate and the total heat released were reduced by 87.7 and 83.5%, respectively, compared with that of the unmodified wood. In addition, the limiting oxygen index reached 32.0%, and the sample passed the UL-94-V-0 rating test. Finally, it was also deduced that the impact strength of the modified wood was superior to that of natural wood. In conclusion, the results showed that the UF-GZ/PA/Cu flame-retardant coating can effectively improve the flame-retardant properties of wood. Importantly, the raw materials employed for the preparation of these flame retardants are abundant and low cost, thereby indicating their potential broad application prospects.

## Figures and Tables

**Figure 1 polymers-16-03366-f001:**
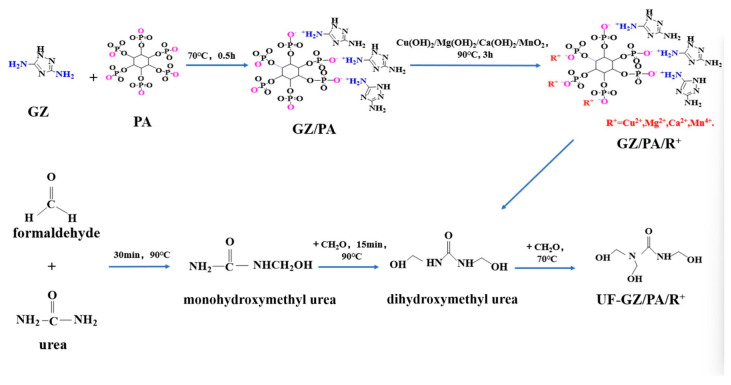
Synthesis of the GZ/PA/metal species.

**Figure 2 polymers-16-03366-f002:**
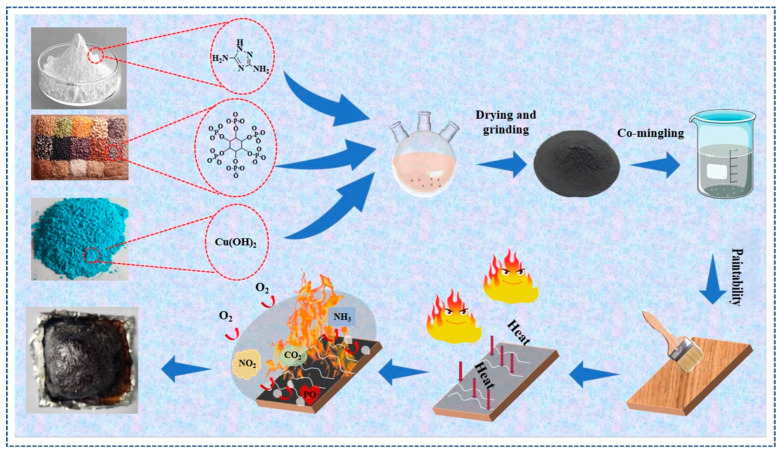
Preparation of the UF-GZ/PA/Cu specimen.

**Figure 3 polymers-16-03366-f003:**
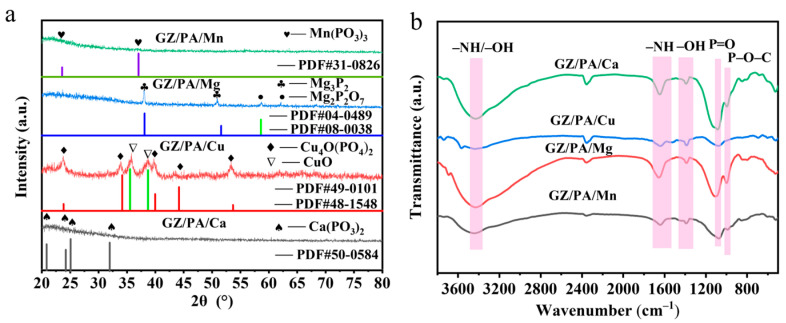
(**a**) XRD results for the prepared GZ/PA/metal species. (**b**) FTIR spectra for the prepared GZ/PA/metal species.

**Figure 4 polymers-16-03366-f004:**
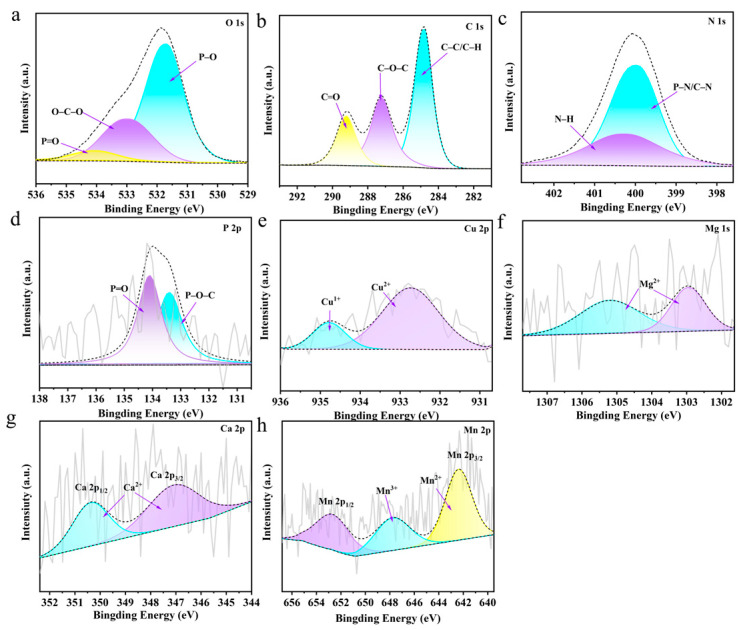
XPS spectra of GZ/PA/Cu (**a**–**e**): (**a**) O 1s, (**b**) C 1s, (**c**) N 1s, (**d**) P 2p, and (**e**) Cu 2p. (**f**) Mg 1s spectrum of GZ/PA/Mg, (**g**) Ca 2p spectrum of GZ/PA/Ca, and (**h**) Mn 2p spectrum of GZ/PA/Mn.

**Figure 5 polymers-16-03366-f005:**
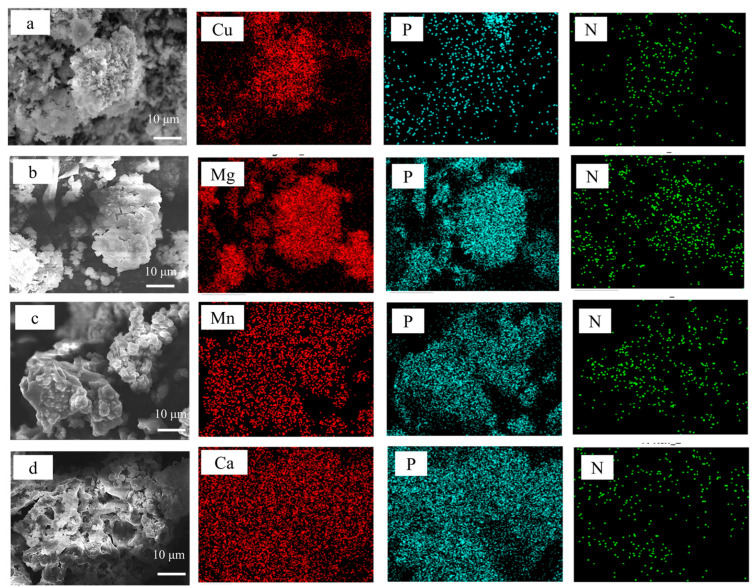
SEM images and EDS maps recorded for (**a**) GZ/PA/Cu, (**b**) GZ/PA/Mg, (**c**) GZ/PA/Mn, and (**d**) GZ/PA/Ca.

**Figure 6 polymers-16-03366-f006:**
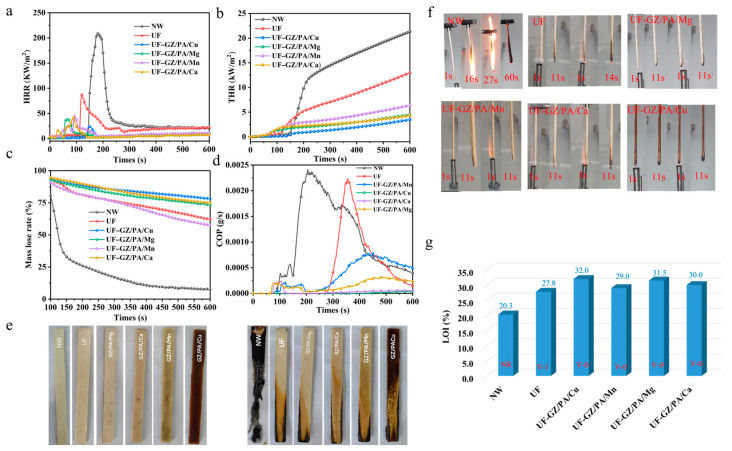
(**a**) HRR curves, (**b**) THR curves, (**c**) mass loss curves, (**d**) COP curves, (**e**,**f**) digital photographic images recorded during UL-94 testing, and (**g**) LOI and UL-94 results for the various specimens.

**Figure 7 polymers-16-03366-f007:**
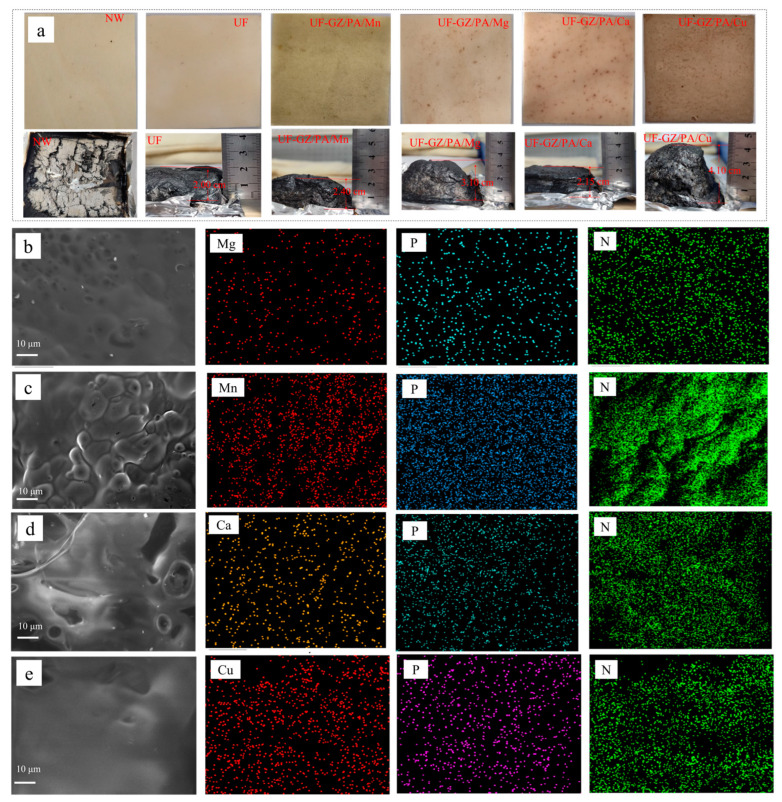
(**a**) Photographic images of the various specimens before and after combustion. SEM and EDS plots are also shown for (**b**) UF-GZ/PA/Mg, (**c**) UF-GZ/PA/Mn, (**d**) UF-GZ/PA/Ca, and (**e**) UF-GZ/PA/Cu after combustion.

**Figure 8 polymers-16-03366-f008:**
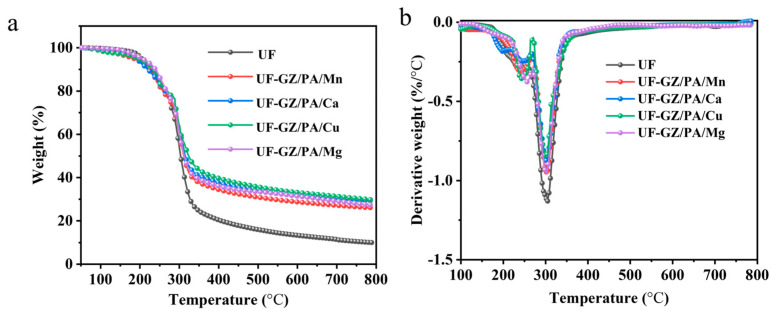
(**a**) TG and (**b**) DTG spectra of the UF and UF-GZ/PA/metal systems.

**Figure 9 polymers-16-03366-f009:**
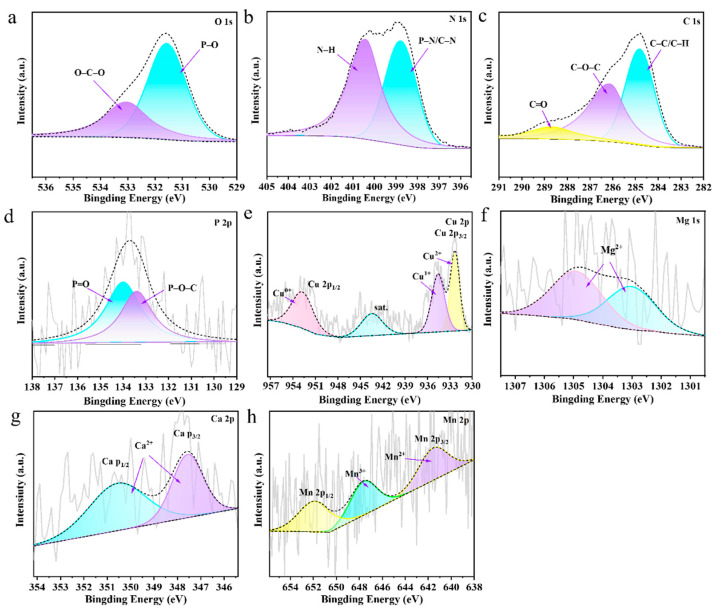
XPS spectra of GZ/PA/Cu (**a**–**e**): (**a**) O 1s, (**b**) C 1s, (**c**) N 1s, (**d**) P 2p, and (**e**) Cu 2p. Also shown are (**f**) the Mg 1s spectrum of GZ/PA/Mg, (**g**) the Ca 2p spectrum of GZ/PA/Ca, and (**h**) the Mn 2p spectrum of GZ/PA/Mn.

**Figure 10 polymers-16-03366-f010:**
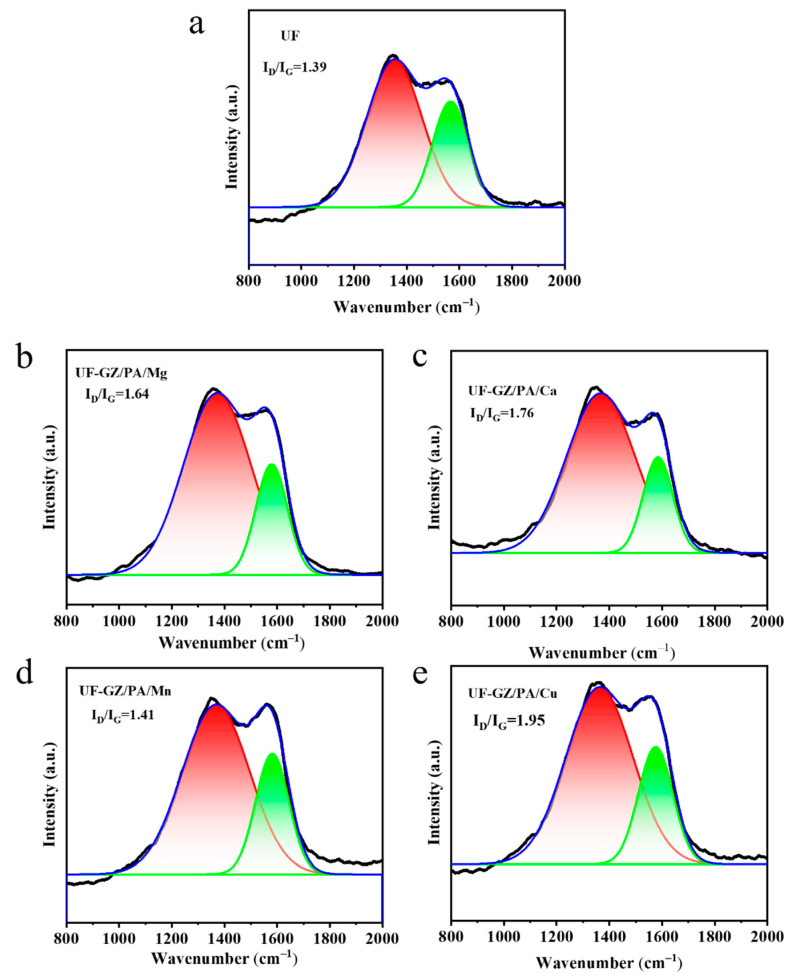
Raman spectra of the residual carbon after combustion for the (**a**) UF, (**b**) UF-GZ/PA/Mg, (**c**) UF-GZ/PA/Ca, (**d**) UF-GZ/PA/Mn, and (**e**) UF-GZ/PA/Cu species.

**Figure 11 polymers-16-03366-f011:**
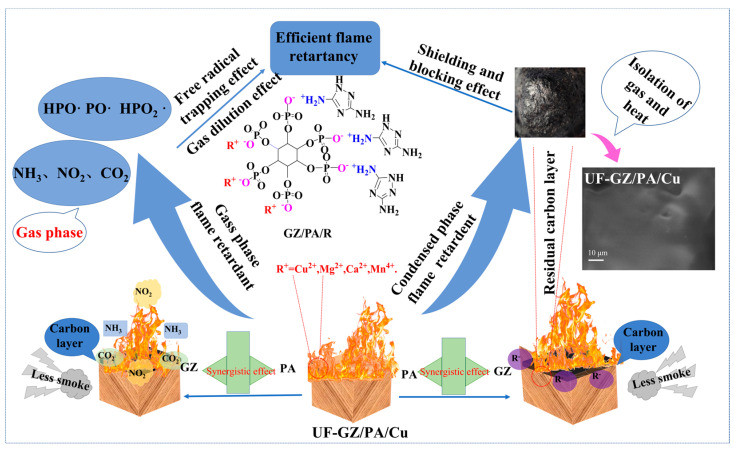
Mechanism of flame retardancy in the UF-GZ/PA/Cu system.

**Figure 12 polymers-16-03366-f012:**
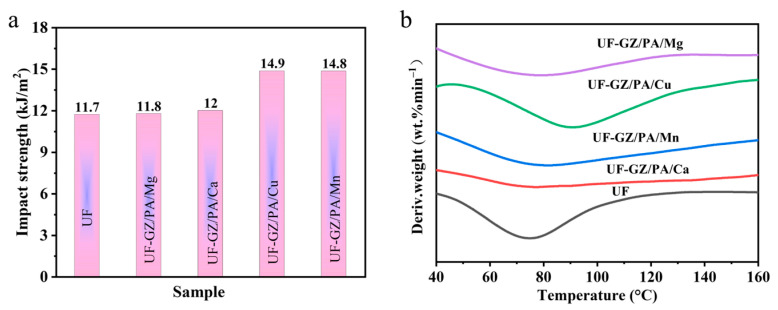
(**a**) Impact strength evaluations for the different resin samples. (**b**) DSC curves for the UF and UF-GZ/PA/metal specimens.

**Table 1 polymers-16-03366-t001:** Vertical combustion test and oxygen index test results for the various flame retardants.

Sample	LOI (%)	T_1_ (s)	T_2_ (s)	Litter	UL-94 Rating
NW	20.3	>60	-	Yes	NR
UF	27.8	10.2	12.6	No	V-1
UF-GZ/PA/Cu	32.0	0.9	1.4	No	V-0
UF-GZ/PA/Mn	29.0	1.7	1.8	No	V-0
UF-GZ/PA/Mg	31.5	1.1	3.9	No	V-0
UF-GZ/PA/Ca	30.0	1.2	2.0	No	V-0

**Table 2 polymers-16-03366-t002:** Cone calorimetry test data for the modified wood samples.

Samples	TTI (s)	PHRR (kW m^−1^)	THR (MJ m^−2^)	av-EHC (MJ kg^−1^)	TML (wt%)	pCOP (g s^−1^)
NW	136	206.73 ± 4.66	21.31	47.26 ± 2.10	92.14 ± 0.53	0.00240 ± 0.00010
UF	136	87.22 ± 3.67	12.98	16.48 ± 1.01	37.76 ± 0.13	0.00220 ± 0.00020
UF-GZ/PA/Cu	133	25.38 ± 2.50	3.51	1.37 ± 2.50	21.81 ± 0.23	0.00006 ± 0.00003
UF-GZ/PA/Mg	70	39.20 ± 3.10	4.49	8.10 ± 0.73	27.65 ± 0.50	0.00030 ± 0.00010
UF-GZ/PA/Mn	95	47.00 ± 2.10	8.66	2.32 ± 3.48	42.67 ± 0.10	0.00070 ± 0.00009
UF-GZ/PA/Ca	83	43.30 ± 1.6	4.35	6.08 ± 1.99	25.15 ± 1.00	0.00020 ± 0.00002

**Table 3 polymers-16-03366-t003:** Comparison of the flame-retardant properties of UF-GZ/PA/Cu flame-retardant coatings with the research data on flame-retardant properties of relevant phytate metal-based flame retardants in recent years.

Samples	Additive Quantity (%)	LOI (%)	PHRR Reduction Rate (%)	THR Reduction Rate (%)	Residual Carbon Rate at 800 °C (%)
UF-GZ/PA/Cu	1	32.0	87.7	83.6	29.8
MPAZn	20	25.0	27.5	21.0	15.0 ± 4.0
Cu(Ⅱ)-PA/WPU-HIMB	-	27.5	≥–3.4	18.0	<8.5

**Table 4 polymers-16-03366-t004:** Summary of the TGA results.

Sample	T_5%_ (°C)	T_max_ (°C)	Char Residue (wt.%)
UF	204.2	304.2	10.0
UF-GZ/PA/Cu	195.5	300.6	29.8
UF-GZ/PA/Mg	205.2	302.8	27.3
UF-GZ/PA/Ca	191.4	301.5	29.4
UF-GZ/PA/Mn	184.0	304.1	26.0

## Data Availability

The original data for this study are included in the article. For any further inquiries, please contact the corresponding author.

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
