# Peer review of "Preparation and Application of a Urea–Formaldehyde-Blended Guanidinium Azole–Phytic Acid–Copper Flame-Retardant Resin Coating"

_polymers, 2024, doi:10.3390/polym16233366_

Round 1

Reviewer 1 Report

Comments and Suggestions for Authors

  Reviewer Comments:

Title of the Manuscript: Preparation and application of a urea–formaldehyde-blended  guanidinium azole–phytic acid–copper flame retardant resin  coating

The manuscript describes the preparation of urea-formaldehyde blended guanidinium azole-phytic acid –copper and its evaluation as flame retardant. The manuscript needs major revision before it may be considered for publication.

1.      Expand the abbreviations when they appear first in the text. Eg: PUC, PBD, etc

2.      The  broad peak in the FTIR spectra at 3434 cm-1 may be due to –OH instead of –NH!

3.      XPS spectra in Fig 3d-h and Fig 8 f, h are highly noisy. Try to record the XPS spectra again.

4.      Fig 3 and Fig 8: Check and correct the spelling of X-axis caption of all the Figures.

5.      Check the peak values of P2p XPS spectrum  mentioned in the text  for residual carbon after combustion!

6.      How come Cuo  is available in Cu2p  XPS spectrum  of residual carbon after combustion as mentioned in the text?

7.      The uniform distribution of GZ/PA/Cu on wood is only the reason for better flame retardancy effect compared to GZ/PA/Mg, GZ/PA/Mn, GZ/PA/Ca! Explain the better flame retardancy effect of GZ/PA/Cu in a more meaningful way.             

Comments on the Quality of English Language

Minor editing of language is required

Author Response

Response to reviewers

We gratefully thank the editor and all reviewers for their time spend making their constructive remarks and useful suggestions, which has significantly raised the quality of the manuscript and has enable us to improve the manuscript. Each suggested revision and comment, brought forward by the reviewers was accurately incorporated and considered. Below the comments of the reviews are response point by point and the revisions are indicated.

Review#1

The manuscript describes the preparation of urea-formaldehyde blended guanidinium azole-phytic acid –copper and its evaluation as flame retardant. The manuscript needs major revision before it may be considered for publication.

  1. Expand the abbreviations when they appear first in the text. Eg: PUC, PBD, etc.

Response: Thanks for your suggestion. In the article PUC is named from the product of the reaction between phytic acid, urea and carboxymethyl lignin, the original article was not clear in my presentation and has been modified accordingly.

  1. The broad peak in the FTIR spectra at 3434 cm-1 may be due to –OH instead of –NH!

Response: Thanks for your suggestion. I have reviewed a lot of literature, and in some literature it says 3000-3500cm-1 for -OH and -NH, and in some literature it says 3000-3500cm-1 for -OH, in addition to 3425cm-1 for -OH and 3439cm-1 for -OH or -NH, for this reason I am going to make the following modifications to the FT-IR graph, and change the 3434cm-1 characteristic peak to -OH/-NH.

  1. XPS spectra in Fig 3d-h and Fig 8 f, h are highly noisy. Try to record the XPS spectra again.

Response: Thank you for your careful review and valuable comments on our work. Regarding your request to re-test the XPS of the samples, we are deeply impressed with the importance of these data for understanding the material properties. However, we regret to inform you that we are unable to provide these data at this time for the following reasons:

â‘  Because the samples have a very low content of metallic elements in the spot range tested, and due to the limitations of the experimental conditions, the XPS test needs to be sent for inspection, which takes a long time, so we do not intend to re-test the XPS.

  1. Fig 3 and Fig 8: Check and correct the spelling of X-axis caption of all the Figures.

Response: Thank you for your suggestions. We have double-checked and corrected the errors in the charts.

  1. Check the peak values of P2p XPS spectrum mentioned in the text for residual carbon after combustion!

Response: Thank you for your comments and valuable comments. From the P2p XPS spectra, it is known that one of the residual char compositions after combustion is composed of pyrophosphates or metaphosphates formed due to the heat of PA.

  1. How come Cuois available in Cu2p XPS spectrum of residual carbon after combustion as mentioned in the text?

Response: Thank you for your suggestions. The reason why Cu0 appears in the Cu2p XPS spectrum of residual carbon after combustion may be due to the thermal oxidative reduction of some of the metallic copper elements to copper monomers.

  1. The uniform distribution of GZ/PA/Cu on wood is only the reason for better flame retardancy effect compared to GZ/PA/Mg, GZ/PA/Mn, GZ/PA/Ca! Explain the better flame retardancy effect of GZ/PA/Cu in a more meaningful way.

Response: Thanks for your suggestion. In GZ/PA/Cu, the dispersion and compatibility of Cu ions in the mechanism is better than that of other metals, Cu ions have high thermal conductivity, and Cu elements show multiple valence states after combustion, which promotes the consumption of oxygen and is conducive to the improvement of flame retardant effect.

Reviewer 2 Report

Comments and Suggestions for Authors

1-      There are some grammatic mistakes. I highly recommend to check the file with a native speaker.

2-      AFM analysis would be highly helpful to characterize the surface.

3-      What is the exact amounts of the metals in the composite? FAA data should be added.

4-      A comparison between EDS and FAA results is necessary to investigate the distribution of the metals in the composite and on the surface.

5-      How the materials can produce lower amount of CO? what is the mechanism behind that? Is there any side-reaction? Site that.

6-      Line 294: which kind of non-combustible gas ? add references.

7-      I highly suggest to compare the data with other some recent research in a table. Also explain what the merit of your research is.

Author Response

  1. There are some grammatic mistakes. I highly recommend to check the file with a native speaker.

Response: Thanks for your suggestion. We have carefully checked and corrected formatting errors in the article.

  1. AFM analysis would be highly helpful to characterize the surface.

Response: Thank you for your careful review and valuable comments on our work. Regarding your request to determine the AFM of the samples, we are deeply impressed by the importance of these data for understanding the surface characteristics of the materials. However, we regret to inform you that we are unable to provide these data at this time due to the following reasons:

We are unable to provide data for AFM testing due to limited access to laboratory instruments.

  1. What is the exact amounts of the metals in the composite? FAA data should be added.

Response: Thank you for your valuable comments and interest in our work. Due to the limitation of experimental conditions and sample production, we regret to inform you that we are unable to provide FAA data.

  1. A comparison between EDS and FAA results is necessary to investigate the distribution of the metals in the composite and on the surface.

Response: Thanks for your valuable comments and attention to our work. We regret to inform you that we are unable to provide FAA data due to limitations in experimental conditions and sample production.

  1. How the materials can produce lower amount of CO? what is the mechanism behind that? Is there any side-reaction? Site that.

Response: Thanks for your in-depth review and valuable comments on our study. When materials are burned at high temperatures, CO is produced due to the reduction of oxygen, which is consumed by the thermal decomposition of PA to form pyrophosphates, the non-combustible gases produced by the heat of GZ, and the oxidation of metals to oxides. The production of large amounts of CO during the combustion of materials is harmful, but the flame retardants synthesized in this work are conducive to reducing the amount of CO produced.

  1. Line 294: which kind of non-combustible gas? add references.

Response: Thank you for your question, the non-flammable gases in this case are NH4, NO or NO2 etc.

  1. I highly suggest to compare the data with other some recent research in a table. Also explain what the merit of your research is.

Response: Thanks for your detailed review and valuable comments on our paper. We research and prepare GZ/PA/Cu flame retardant, add 1% of the amount, that is, add a small amount of flame retardant can be obtained better flame retardant effect, and the product of the raw material source is abundant, low cost, and is conducive to the realization of the factory.

Table 1 Comparison table of the flame retardant properties of some latest research PA/Cu flame retardants and this solution UF-GZ/PA/Cu

Sample

additive quantity (%)

LOI (%)

PHRR reduction rate (%)

THR (%)

Carbon Residual Rate (%)

UF-GZ/PA/Cu

1%

32.0

87.7

83.6

29.8

Cu(II)–PA/WPU–HIMB

-

27.5

-4.0

9.9

8.5

PA-Cu-CPCC

-

33.8

13.1

26.6

-

PA/CH-U-Cu

12%

24.5

50

-

25.5

Reviewer 3 Report

Comments and Suggestions for Authors

1. Abstract is incomplete, rephrase and all comprehensive results in this section.

2. Materials and methodology section should contain attractive figure showing all the synthesis steps.

3. FTIR results are poor written, it should contain information of all characteristics peaks.

4. Author should explain this statement with respect to literature review " while the UF coating itself exhibits a degree of flame retardancy, it generates a fragmented residual carbon layer of poor quality, which limits the flame retardancy of the system".

5. Conclusion is incomplete.

Comments on the Quality of English Language

English quality must be improved

Author Response

  1. Abstract is incomplete, rephrase and all comprehensive results in this section.

Response: Thank you for reviewing our paper and for your valuable comments. We have reworded the description of the abstract accordingly.

  1. Materials and methodology section should contain attractive figure showing all the synthesis steps.

Response: Thank you for your advice. We have drawn a flowchart that includes all the steps of the experiment.

  1. FTIR results are poor written, it should contain information of all characteristics peaks.

Response: Thank you for your valuable comments and interest in our work. We have re-described and re-analyzed the FT-IR data.

  1. Author should explain this statement with respect to literature review " while the UF coating itself exhibits a degree of flame retardancy, it generates a fragmented residual carbon layer of poor quality, which limits the flame retardancy of the system".

Response: Thanks for your suggestion. Because UF has a high ratio of C and H elements and is prone to carbon fragmentation.

  1. Conclusion is incomplete.

Response: Thank you for your suggestions. We have revised our conclusions.

Round 2

Reviewer 1 Report

Comments and Suggestions for Authors

Manuscript may be considered for publication

Comments on the Quality of English Language

Quality of English language seems to be ok

Author Response

The comments have been responsed.

Reviewer 3 Report

Comments and Suggestions for Authors

Comments are addressed. Can be accepted in current form. 

Author Response

The comments haved responsed in the revised manuscript.